# Function of *crzA* in Fungal Development and Aflatoxin Production in *Aspergillus flavus*

**DOI:** 10.3390/toxins11100567

**Published:** 2019-09-27

**Authors:** Su-Yeon Lim, Ye-Eun Son, Dong-Hyun Lee, Tae-Jin Eom, Min-Ju Kim, Hee-Soo Park

**Affiliations:** School of Food Science and Biotechnology, Institute of Agricultural Science and Technology, Kyungpook National University, Daegu 41566, Korea; syl_253@naver.com (S.-Y.L.); thsdpdms0407@naver.com (Y.-E.S.); lkj20107@naver.com (D.-H.L.); utj0225@naver.com (T.-J.E.); 13mjkim@gmail.com (M.-J.K.)

**Keywords:** CrzA, Calcineurin, aflatoxin, *Aspergillus flavus*, Asexual development

## Abstract

The calcineurin pathway is an important signaling cascade for growth, sexual development, stress response, and pathogenicity in fungi. In this study, we investigated the function of CrzA, a key transcription factor of the calcineurin pathway, in an aflatoxin-producing fungus *Aspergillus flavus (A. flavus)*. To examine the role of the *crzA* gene, *crzA* deletion mutant strains in *A. flavus* were constructed and their phenotypes, including fungal growth, spore formation, and sclerotial formation, were examined. Absence of *crzA* results in decreased colony growth, the number of conidia, and sclerocia production. The *crzA*-deficient mutant strains were more susceptible to osmotic pressure and cell wall stress than control or complemented strains. Moreover, deletion of *crzA* results in a reduction in aflatoxin production. Taken together, these results demonstrate that CrzA is important for differentiation and mycotoxin production in *A. flavus*.

## 1. Introduction

*Aspergillus flavus* is widespread in our environment and contaminates crops such as maize, corn, and peanut [1,2,3]. *A. flavus* spores germinate in crops and produce detrimental secondary metabolite mycotoxins, including aflatoxins [4,5]. Aflatoxins are harmful fungal mycotoxins that cause carcinogenesis in animals and humans [6,7]. *A. flavus* is also an opportunistic pathogen in humans and is the second leading pathogen of invasive aspergillosis [8,9,10]. Because of the harmful effects of this fungus, *A. flavus* causes economic loss to humankind, and developing a better understanding of the biology of this fungus is vital to reduce the adverse effects of *A. flavus* on humanity.

*A. flavus* is a heterothallic fungus that can reproduce both sexually and asexually [11,12,13]. Sclerotia are sexual reproductive structures that can be produced by a single strain of one mating type or a cross between opposite mating type strains [14]. Sclerotia can be widespread in soil or crops and survive environmental stresses [14]. *A. flavus* also produces asexual spores, called conidia, that act as propagates and are infectious [1]. Asexual development is the primary reproductive mode in most *Aspergillus* species [15]. *A. flavus* produces the asexual-specific structure, conidiophores, during asexual development. The production of conidiophores is tightly controlled by a variety of positive and negative regulators [16]. Both asexual and sexual development is regulated by several signaling pathways in *Aspergillus* species [17,18].

The Calcium-calcineurin signaling pathway is highly conserved and plays key roles in fungi, including adaptation to hosts or environments, fungal growth, pathogenesis, and sexual reproduction [19,20]. Calcium is a key ion for signal transduction in response to various environmental stresses [21,22]. During stress, intracellular calcium concentration increases, and then calcium ions in cytoplasm bind to calmodulin. The calcium-calmodulin complex activates the calmodulin-dependent phosphatase calcineurin [23]. Calcineurin dephosphorylates certain targets that function to control the stress response, growth, fungal virulence, and pathogenesis in fungi [24,25]. In fungi, one of the key target proteins of calcineurin is a C_2_H_2_ transcription factor Crz1 (or CrzA, calcineurin responsive zinc finger 1) [26]. Activated Crz1 regulates certain genes involved in cell wall integrity and virulence [26]. A variety of studies found that Crz1 plays vital roles in ion homeostasis, cell wall biogenesis, fungal development, stress response, and pathogenesis [26]. In *Saccharomyces cerevisiae*, loss of *CRZ1* results in alteration of mRNA expression of genes involved in cell wall biogenesis, ion homeostasis, and vesicle transport, suggesting that Crz1 is crucial for cell wall biogenesis and ion homeostasis [27,28]. In human pathogenic fungi such as *Cryptococcus neoformans* and *Candida albicans*, the absence of *CRZ1* attenuates virulence in mouse models [29,30,31]. Crz1 is also involved in fungal pathogenesis in plant pathogenic fungi *Magnaporthe oryzae* and *Botrytis cinerea* [32,33]. In *Aspergillus* species, CrzA is also required for fungal growth, asexual development, and septation [19,34,35]. Absence of *crzA* results in reduced hyphal growth and asexual development in *Aspergillus nidulans* [36,37,38]. CrzA is also required for aflatoxin production in *A. parasiticus* [39]. In *A. fumigatus*, Δ*crzA* mutant strains exhibit abnormal hyphae morphology and septation, reduced asexual sporulation, attenuated virulence, and increased susceptibility to ionic stresses [34,35]. Genome-wide analyses of *crzA* in *A. fumigatus* demonstrated that CrzA regulates gene expression associated with ion transport, secondary metabolism, lipid metabolism, and vesicle trafficking [40,41]. These results strongly suggest that CrzA plays diverse roles in *Aspergillus* species [19,26]. Although CrzA plays important roles in *Aspergillus* species, these roles have not been demonstrated in *A. flavus*. In this study, we characterized CrzA in the aflatoxin-producing fungi *A. flavus*. To examine the role of CrzA, *crzA* deletion mutant (Δ*crzA*) strains were generated, and their phenotypes were analyzed. Similar to the CrzA role in other *Aspergillus* species, our results demonstrate that CrzA is essential for fungal growth, asexual spore formation, stress response, and aflatoxin production in *A. flavus.*

## 2. Results and Discussion

### 2.1. Summary of CrzA in A. flavus

Crz1 is a C_2_H_2_-type transcription factor that is a key target for calcineurin in yeast and filamentous fungi [26]. Recent studies analyzed the function and domains of CrzA in two *Aspergillus* species *A. fumigatus* and *A. nidulans* [42,43]. CrzA from both species contains putative a calcineurin-binding domain (CBD), DNA-binding domain (DBD, C_2_H_2_ domain), nuclear localization signal (NLS), and serine-rich region. Although *A. fumigatus* CrzA contains a nuclear export signal (NES) at its N-terminus, *A. nidulans* CrzA has no NES at its N-terminus. To identify CrzA in *A. flavus*, we screened the *A. flavus* NRRL 3357 genome [44] using the protein sequence of *A. fumigatus* CrzA (XP_750439) and *A. nidulans* CrzA (XP_663330.1). XP_002381985.1 was identified by protein homology (69% and 67% identity with *A. fumigatus* CrzA and *A. nidulans* CrzA, respectively) (Figure 1). AFLA_127920, a putative *crzA* gene, encodes a 773-amino acid protein that contains a DNA binding domain with two C_2_H_2_ Zn-finger domains at the C-terminus. CrzA also has a calcineurin binding domain with calcineurin docking sequence (PxIxT) [45]. The *A. flavus* CrzA protein contains a putative NLS element (nls-mapper.iab.keio.ac.jp) [45]. However, *A. flavus* CrzA does not contain an NES at the N-terminus, unlike the *A. fumigatus* CrzA. The *A. flavus* CrzA protein also contains CBD and serine-rich regions that are essential for regulating the activity of CrzA [42,43]. The DBD, CBD, and NLS elements of the CrzA homologs in most *Aspergillus* spp are highly conserved, but the NES element is diverse in 19 *Aspergillus* species (Figure 1). These results suggest that AFLA_127920 encodes the CrzA homolog of *A. flavus*.

### 2.2. Roles of CrzA in Asexual Development

To test the functions of CrzA in *A. flavus*, the *crzA* deletion (∆*crzA*) mutants and complemented strains (C’ *crzA*) were generated and compared the phenotype of the ∆*crzA* mutant with the control strain and the complemented strain. Growth and conidiation under light and dark conditions for wild type (WT) and ∆*crzA* mutant strains were evaluated (Figure 2A,B). The colony diameter of the Δ*crzA* strains was significantly decreased compared with control and the complemented strains (Figure 2C). The number of asexual spores in Δ*crzA* strains was lower in both light and dark conditions (Figure 2D). The Δ*crzA* strains produce light green or brown conidiophores, while control and complemented strains produce green conidiophores (Figure 2A). Microscopic analysis exhibited that the Δ*crzA* strains produce abnormal conidiophores in the light condition (Figure 2E). Overall, these results suggest that CrzA is not only necessary for fungal growth but also play a critical role in the formation of conidiophores in *A. flavus*.

Because CrzA plays an important role in conidiophore production, mRNA expression of genes that regulate asexual development including *brlA*, *abaA*, and *wetA* was assessed in WT and the Δ*crzA* mutant strains. We first examined the mRNA expression of *brlA*, a key initiator of conidiation, during the early phase of conidiation (after asexual induction 12–24 h). The *brlA* mRNA level was lower in the Δ*crzA* mutant compared to control and complemented strains (Figure 3A). Similarly, *abaA* and *wetA* mRNA levels in the Δ*crzA* mutant were lower compared to those in control and complemented strains (Figure 3B,C). These results demonstrate that CrzA is crucial for the proper expression of key regulators of asexual development.

As mention above, CrzA is required for proper the morphology and production of conidiophores. The color of conidiophores of the Δ*crzA* mutant strain was changed to brownish in both dark and light conditions (Figure 2A,B). The size of conidiophore heads and the number of conidia in the Δ*crzA* mutant strains were each decreased compared to control strains (Figure 2D,E). In addition, the Δ*crzA* mutant strains produce less amount of conidia compare to control strains. The phenotypes of Δ*crzA* mutant in other *Aspergillus* species *A. nidulans* [37] and *A. fumigatus* [34] are similar to those of the Δ*crzA* mutant in *A. flavus*, implying that CrzA play a conserved role in asexual development in *Aspergillus* species. Absence of *crzA* decreased the mRNA expression of *brlA*, *abaA*, and *wetA* (Figure 3). In the model organism *A. nidulans* and the pathogenic fungus *A. fumigatus*, *brlA* expression is dependent on CrzA during asexual development [34]. These results suggest that the altered *brlA* mRNA expression by deletion of *crzA* affects morphogenesis of conidiophores in *Aspergillus* species.

### 2.3. Roles of CrzA in Sclerocia Formation

Because CrzA in *A. parasiticus* is crucial for sclerotial development [39], we examined the role of *crzA* in sclerocia formation in *A. flavus*. Control and mutant strains were inoculated into glucose minimal medium with 0.1% yeast extract (MMYE) media and cultured for 7 days under dark conditions. While WT and control and complemented strains produce conidiophores and sclerocia, the Δ*crzA* strains do not produce developmental structures under dark conditions, suggesting that CrzA is essential for proper sclerocia production in *A. flavus* (Figure 4).

### 2.4. Roles of CrzA in Stress Response

CrzA homologs are involved in cation homeostasis and stress response in *A. nidulans* and *A. fumigatus* [34,35,37,46]. Control and mutant strains were inoculated in MMYE with several compounds that are related with osmotic stress (sorbitol, NaCl, and KCl), ion homeostasis (CaCl_2_), and cell wall stress (calcofluor white (CFW) and Congo red (CR)) (Figure 5). The Δ*crzA* mutant strains exhibit reduced colony growth or fungal development under all conditions tested. Importantly, Δ*crzA* mutant strains cannot grow at high concentrations of Ca^2+^ ions. However, Δ*crzA* mutant strains are not sensitive to oxidative stresses (1–5 mM H_2_O_2_, data not shown). We also examined the thermo-tolerance of Δ*crzA* mutant strains, but the thermal sensitivity of Δ*crzA* mutant strains was similar to WT. We then checked the contents of trehalose, a protectant for several stresses, in mutant conidia [47]. However, trehalose content of the Δ*crzA* mutant conidia was the same as that in WT conidia (Appendix A). This result suggests that CrzA may not be involved in trehalose biosynthesis. Overall, these results demonstrated that the growth of Δ*crzA* mutant strains was reduced in various stress conditions in *A. flavus* (Figure 5), implying that the role of CrzA in cell wall integrity and the stress tolerance is conserved in most filamentous fungi.

Previous results from genome-wide analyses found CrzA governs mRNA expression of certain genes associated with calcium metabolism, ion transport, and cell wall integrity [34,36]. Several genes contain CrzA-binding sites (G[A/T]GGC[G/C]) in their promoter regions [41,48,49], thus, we measured the mRNA levels of putative target genes, including *rcnA*, *chsB*, *pmrA*, and *pmcA* (Appendix A). Under normal conditions, the mRNA expression of these three genes decreased in the Δ*crzA* mutant strains in liquid culture. We then added calcium ions to the liquid media and measured mRNA expression. The mRNA levels of *rcnA* and *pmcA*, but not *chsB* were increased in response to high calcium concentration media in WT. However, the mRNA levels of *rcnA* and *pmcA* were dramatically decreased in the Δ*crzA* mutant strains. While the expression levels of three genes *rcnA*, *chsB*, and *pmcA* were decreased in the Δ*crzA* mutant than those in control, *pmrA* mRNA level was not changed in the Δ*crzA* mutant. These results indicate that CrzA supports *rcnA* and *pmcA* expression and this activity is conserved in other *Aspergillus* species [34,43]. Overall, these finding indicate that the roles of CrzA in cell wall biogenesis and stress response are conserved in *Aspergillus* species, but target genes of CrzA are diverse in *Aspergillus* species. Additional genome-wide analyses in *A. flavus* will provide insight into the role of CrzA in *Aspergillus* species.

### 2.5. Roles of CrzA in Aflatoxin B1 Production

Because CrzA is essential for aflatoxin B1 production in *A. parasiticus* [39], we examined aflatoxin production in *A. flavus*. Control, Δ*crzA,* and C’*crzA* strains were inoculated into liquid media and incubated for 7 days under the dark condition. As shown Figure 6, control and complemented strains produce aflatoxin B1, whereas Δ*crzA* strains do not produce aflatoxin B1, suggesting that CrzA is required for aflatoxin B1 production. The spot pattern of Δ*crzA* mutant on TLC plates was different from that of control strains, speculating that deletion of *crzA* can alter secondary metabolite biosynthesis (Figure 6). Further high-performance liquid chromatography (HPLC) or other analytic analyses will help to understand the role of CrzA in secondary metabolite biosynthesis.

### 2.6. Roles of CrzA in Host Colonization

We then examined whether loss of *crzA* also affects host colonization. Control, Δ*crzA* mutant, and complemented strains were inoculated in kernels and incubated for 5 days under light conditions. During conidial production, Δ*crzA* mutant strains exhibit a lower sporulation level compared to control and complemented strains (Figure 7).

## 3. Conclusions

CrzA is a putative calcineurin target that coordinates fungal growth and asexual or sexual reproduction in *A. flavus*. The role of CrzA in stress response and cell wall integrity is conserved in *Aspergillus* species. Importantly, CrzA is crucial for aflatoxin production and host infection. Taken together, we propose that inhibition of calcineurin and CrzA could inhibit survival and aflatoxin production in a variety of environmental conditions and that calcineurin could be an attractive target for anti-fungal agents.

## 4. Materials and Methods

### 4.1. Fungal Strains and Culture Conditions

Table 1 shows a list of strains used in this study. To culture *A. flavus* strains, glucose minimal medium with 0.1% yeast extract (MMYE) was used [50]. *A. flavus* NRRL 3357.5 (*pyrG*^-^ auxotrophic mutant strain) was used for transformation and was grown on MMYE with supplements (5 mM uridine (Acros organics, NJ, USA) and 10 mM uracil (Acros organics, NJ, USA)) [51].

### 4.2. Generation of the CrzA Mutant Strains

The primers used in this study are listed in Table 2. The double-joint PCR (DJ-PCR) method was used for generation of the *crzA* deletion mutant strain [52]. The 5′ and 3′ flanking regions of the *crzA* gene were both amplified from *A. flavus* NRRL 3357 genomic DNA using OHS357:OHS358 and OHS359:OHS360, respectively. The *A. fumigatus pyrG*^+^ marker was amplified with the primer pair OHS089:OHS090 from *A. fumigatus* AF293 genomic DNA. The *crzA* deletion cassette was amplified with OHS361:OHS362 and introduced into protoplasts of the recipient strain *A. flavus* NRRL 3357.5. Protoplasts were generated using the VinoTaste^®^ Pro (Novozymes, Denmark) and lysing enzyme (Sigma, St Louis, MO, USA) [53]. Genomic DNA of transformants was isolated and the coding region of *crzA* was amplified. PCR fragments were digested by several restriction enzyme to verity deletion mutants. Three independent Δ*crzA* strains (TDH1.1-3) were isolated.

For complemented *crzA* strains, the promoter and ORF regions of *crzA* was amplified with the primer pair OHS367/OHS368, purified, digested with *Not*I, and cloned into pMJ4 containing a 4xFLAG tag, the pyrithiamine-resistance gene (*ptrA*), and the *amyB* terminator. Then, the resulting plasmid pTJ5 was introduced into the recipient ∆*crzA* strain TDH1.1 to give rise to TTJ9.1-2. Transformants were verified by qRT-PCR (Appendix A).

### 4.3. Physiological Studies

To determine the number of conidia, control and mutant strains were point-inoculated onto solid MMYE and incubated at 30 °C for 5 days under dark or light conditions. Colony diameter for each strain was then measured for fungal growth. For conidial production, conidia from each strain were collected from the entire plate and counted using a hemocytometer. For sclerocia production, fungal strains were point-inoculated onto solid MMYE and incubated at 30 °C for 7 days under dark conditions. The cultured plates were then washed with 70% ethanol to facilitate sclerotia counting. All experiments were carried out in triplicate.

### 4.4. Quantitative Reverse Transcript PCR (qRT-PCR)

Samples were collected as previously described [54,55]. WT and mutants conidia were inoculated into 500 mL of liquid MMYE and incubated at 37 °C and 200 rpm for 16 h. The mycelia were collected by filtering through autoclaved Miracloth (Calbiochem, CA, USA), mono-layered on solid MMYE, and then incubated at 37 °C under light conditions. For RNA extraction, individual samples were collected at a designated time point.

Total RNA isolation was carried out as previously described [56]. Briefly, each sample was mixed with TRIzol Reagent (Invitrogen, USA) and 0.5 mm Zirconia/Silica beads (RPI, IL, USA), and then homogenized using a Mini-Beadbeater (BioSpec Products Inc, OK, USA). After homogenization, the aqueous phase was separated by centrifugation and transferred to new vials. An equal volume of 70% ethanol was added into the aqueous phase and mixed gently. After centrifugation, RNA was collected. To remove genomic contamination, each sample was treated with DNase I (Promega, WI, USA) and pure RNA was isolated using RNeasy mini Kit (Qiagen Inc., Valencia, USA). The RNA concentration was measured by ultraviolet (UV) spectrometry.

To synthesize complementary DNA (cDNA), the GoScript Reverse Transcription system (Promega, WI, USA) was used. Quantitative PCR was performed with iTaq universal SYBR Green supermix (Bio-Rad, CA, USA) and each gene-specific primer set using a CFX96 Touch Real-Time PCR system (Bio-Rad, CA, USA). To calculate the expression levels of target genes, the 2^−∆∆CT^ method was used. For endogenous control, β-actin gene was used. Results shown in this study are representative of two independent experimental replicates. The primers used for quantitative PCR are listed in Table 2.

### 4.5. Stress Tests

To examine susceptibility to various stresses, 10^3^–10^4^ conidia were inoculated onto MMYE agar media containing the following compounds; Congo red (CR, Sigma, St. Louis, MO, USA) and calcofluor white (CFW, Sigma) for cell wall stress, CaCl_2_ (Sigma) for ion stress, and NaCl (Fisher, Fair Lawn, NJ, USA), KCl (Fisher), and Sorbitol (Fisher) for osmotic stress. The plates were incubated at 30 °C and photographed between 3 and 5 days post-treatment.

### 4.6. Aflatoxin Extraction and Thin-Layer Chromatography Analysis

Aflatoxin B1 from each sample was extracted as described by previously [56], with modification. Briefly, approximately 10^6^ spores of each strain were inoculated into 5 mL of liquid complete medium (CM) and cultured for 7 d at 30 °C under dark conditions. To extract Aflatoxin B1, CHCl_3_ (5 mL) was added to the cultured media and mixed vigorously by a Voltex mixer. The mixed sample was centrifuged and the organic phase was transferred to new vials. Sample was evaporated and resuspended in 50 μL of CHCl_3_ for TLC (Thin layer chromatography) analysis. Suspended solutions containing Aflatoxin B1 were spotted onto a TLC silica plate (Kiesel gel 60, 0.25 mm; Merck). The plate was putted into a chamber containing chloroform: acetone (9:1, v/v). To capture the image, the TLC plates were exposure to ultra violet (UV) light (366 and 254 nm).

### 4.7. Kernel Bioassays

Kernel bioassays were carried out as described previously [57]. Corn kernels (Chungnong, Bucheon, South Korea) were washed in 70% ethanol, rinsed with ddH_2_O, and incubated in bleach. After then, the kernels were washed using ddH_2_O to remove the bleach, and then dried on paper towels. A hole in the embryo of each kernel was made using a sterile needle. Kernels were put into a new vial and inoculated with a conidial suspension (2 × 10^5^ conidia/mL). After inoculation, vials were placed in a 37 °C incubator under light conditions for 5 days. After the 5-day incubation with light, 2 mL of 0.01% Tween 20 was added to each vial, and then 1 mL of conidial suspension was removed from the vials to count the number of conidia using a hemocytometer.

### 4.8. Microscopy

Colony photographs were taken with a Pentax MX-1 digital camera. Microscopic images were taken using a Zeiss Lab.A1 microscope equipped with AxioCam 105 and AxioVision (Rel. 4.9) digital imaging software.

### 4.9. Statistical Analysis

For statistical analysis, GraphPad Prism Version 5.01 software was used. Statistical differences between control and mutant strains were analyzed by the Student’s unpaired *t*-test. Error bars indicates the standard errors from the mean of triplicates. P values < 0.05 were considered to be significant different between control and Δ*crzA* mutant strain. (*, *p* ≤ 0.05; **, *p* ≤ 0.01; ***, *p* ≤ 0.001).

## Figures and Tables

**Figure 1 toxins-11-00567-f001:**
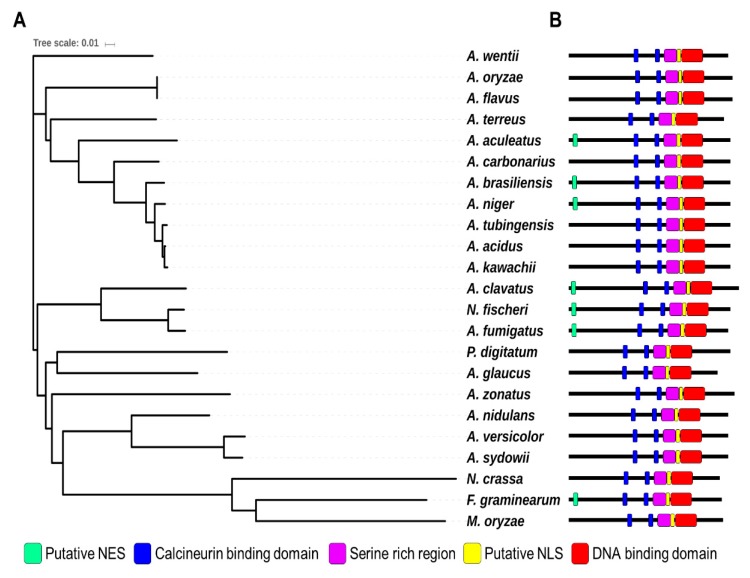
Summary of CrzA in *Aspergillus* species. (**A**) A phylogenetic tree of the CrzA homologs identified in 23 fungal species including *A. flavus* AFL3357 (AFL2T_09134), *A. oryzae* RIB40 (AO090001000491), *A. fumigatus* Af293 (Afu1g06900), *A. nidulans* FGSC4 (AN5726), *A. niger* CBS 513.88 (An18g05920), *A. clavatus* NRRL 1 (ACLA_027670), *A. sydowii* (Aspsy1_0149570), *A. tubingensis* CBS 134.48 (Asptu1_0114323), *A. acidus* CBS 106 47 (Aspfo1_0139906), *A. brasiliensis* CBS 101740 (Aspbr1_0197410), *A. versicolor* CBS 583.65 (Aspve1_0132792), *A. kawachii* (Aspka1_0180272), *A. carbonarius* ITEM 5010 (Acar5010_128721), *A. aculeatus* ATCC16872 (Aacu16872_028665), *A. terreus* NIH2624 (ATET_02928), *A. wentii* DTO 134E9 (Aspwe1_0050154), *A. zonatus* (Aspzo1_0014943), *A. glaucus* CBS 516 65 (Aspgl1_0031266), *Neosartorya fischeri* NRRL 181 (NFIA_017790), *Penicillium digitatum* Pd1 (XP_014531047.1), *Neurospora crassa* OR74A (XP_962085), *Magnaporthe oryzae* 70-15 (MGG_05133), and *Fusarium graminearum* PH-1 (FGSG_01341). A phylogenetic tree of CrzA homologs (or orthologs) was generated by the Clustal Omega package (https://www.ebi.ac.uk/Tools/msa/clustalo/). The tree result was submitted to iTOL (http://itol.embl.de/) to generate the figure. (**B**) Domains of the CrzA homologs in fungal species. NLS (Nuclear Localization Sequence) and NES (Nuclear Export Signal)

**Figure 2 toxins-11-00567-f002:**
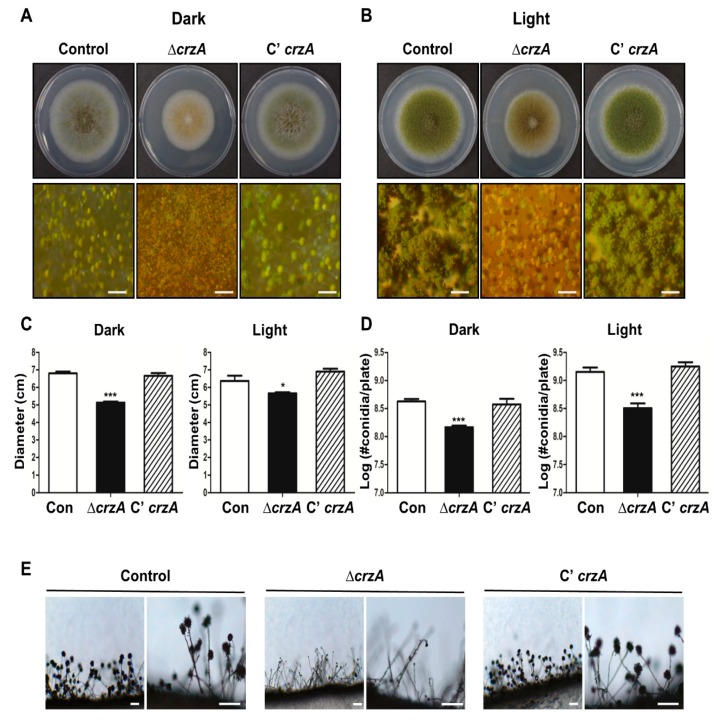
Phenotypes of the Δ*crzA* mutant. (**A**,**B**) Colony phenotypes of control (TTJ6.1), Δ*crzA* (TDH1.1), and C’*crzA* (TTJ9.1) strains point-inoculated on solid glucose minimal medium with 0.1% yeast extract (MMYE) media and grown at 37 °C under dark (**A**) or light (**B**) condition for 5 days (Scale bars = 500 μm). Quantitative analysis of colony diameter (**C**) and the number of conidia per plate (**D**) of WT and velvet deletion mutant strains are shown in (**A**,**B**). Error bars indicates the standard errors from the mean of triplicates. (Control versus Δ*crzA* strains, * *p* ≤ 0.05; *** *p* ≤ 0.001). (**E**) Conidia formation of control (TTJ6.1), Δ*crzA* (TDH1.1), and C’*crzA* (TTJ9.1) strains was observed under a light microscope at 48 h after inoculation onto solid MMYE media at 37 °C. (Scale bars = 250 μm)

**Figure 3 toxins-11-00567-f003:**
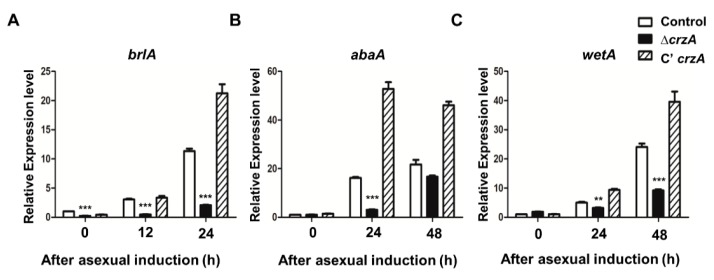
mRNA expression of asexual developmental genes in the Δ*crzA* mutant. (**A**–**C**) mRNA levels of key regulator genes *brlA* (**A**), *abaA* (**B**), and *wetA* (**C**), in control (TTJ6.1), Δ*crzA* (TDH1.1), and C’*crzA* (TTJ9.1) strains after induction of asexual development were assessed by quantitative reverse transcript PCR (qRT-PCR). Error bars indicates the standard errors from the mean of triplicates. (Control versus Δ*crzA* strains, ** *p* ≤ 0.01; *** *p* ≤ 0.001).

**Figure 4 toxins-11-00567-f004:**
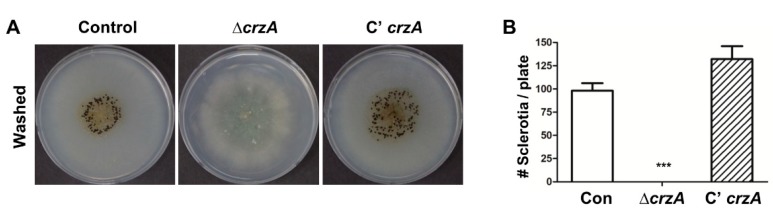
Sclerocia production in the Δ*crzA* mutant. Ethanol-washed colony photographs of control (TTJ6.1), Δ*crzA* (TDH1.1), and C’*crzA* (TTJ9.1) strains grown on solid MMYE media for 7 days. (**B**) Quantitative analysis of sclerocia of strains shown in (**A**). (Control versus Δ*crzA* strains, *** *p* ≤ 0.001).

**Figure 5 toxins-11-00567-f005:**
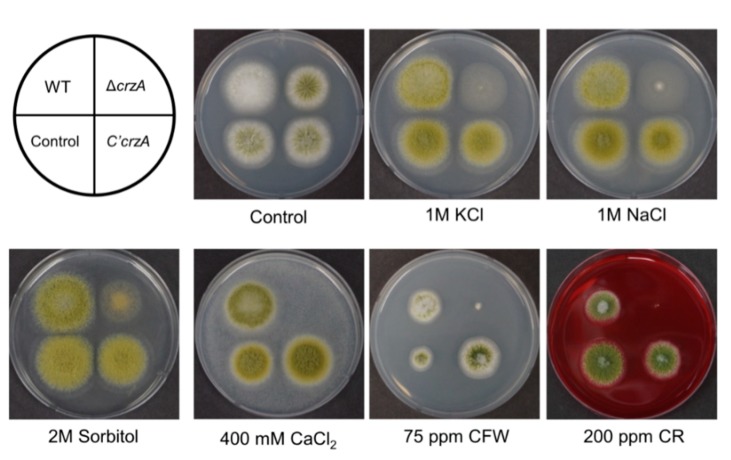
Phenotypes of the Δ*crzA* mutant in various stress conditions. WT (NRRL3357), control (TTJ6.1), Δ*crzA* (TDH1.1), and C’*crzA* (TTJ9.1) strains were point inoculated at 37 °C for 3 days on solid MMYE medium containing various compounds including KCl, NaCl, sorbitol, CaCl_2_, Calcoflour white (CFW), and Congo red (CR).

**Figure 6 toxins-11-00567-f006:**
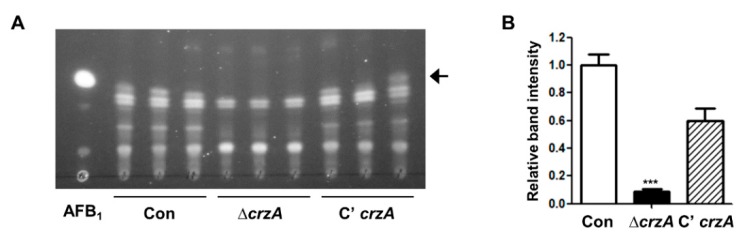
Aflatoxin production in the Δ*crzA* mutant. (**A**) Thin-layer chromatography (TLC) of aflatoxin B1 (AFB1) from control (TTJ6.1), Δ*crzA* (TDH1.1), and C’*crzA* (TTJ9.1) strains under dark conditions for 7 days. Black arrow indicates aflatoxin B1. (**B**) Densitometry of the TLC analysis results is shown in (A). (Control versus Δ*crzA* strains, *** *p* ≤ 0.001).

**Figure 7 toxins-11-00567-f007:**
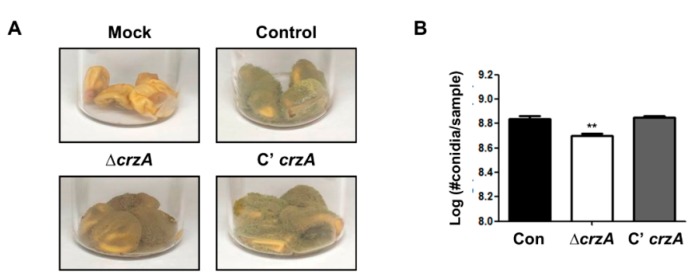
Kernel assay in the δ*crza* mutant. (**A**) Infected kernels were co-incubated with control (TTJ6.1), Δ*crzA* (TDH1.1), and C’*crzA* (TTJ9.1) strains for 5 days. (**B**) The number of conidia per samples (**A**). (Control versus Δ*crzA* strains, ** *p* ≤ 0.01).

**Table 1 toxins-11-00567-t001:** *Aspergillus flavus* strains used in this study.

Strain Name	Relevant Genotype	References
NRRL3357	*A. flavus* wild type	FGSC ^1^
NRRL3357.5	*pyrG* ^-^	(He et al. 2007)
TTJ6.1	*pyrG*^-^; Δ*pyrG*::*AfupyrG^+^*	This study
TDH1.1-3	*pyrG*^-^; Δ*crzA*::*AfupyrG*	This study
TTJ9.1-2	*pyrG*^-^; Δ*crzA*::*AfupyrG*; *crzA(p)*::*crzA*::*FLAG_4x_*::*ptrA*	This study

^1^ Fungal Genetic Stock Center.

**Table 2 toxins-11-00567-t002:** Oligonucleotides used in this study.

Name	Sequence (5′ → 3′) ^a^	Purpose
OHS089	GCTGAAGTCATGATACAGGCCAAA	*AfupyrG* Maker_F
OHS090	ATCGTCGGGAGGTATTGTCGTCAC	*AfupyrG* Maker_R
OHS357	GTTGTCGTCAGGCACCGTCA	5′ flanking region of *crzA*
OHS358	*GGCTTTGGCCTGTATCATGACTTCA* GATACCACGCGAAGCAAGGC	*crzA* with *AfupyrG* tail R
OHS359	*TTTGGTGACGACAATACCTCCCGAC* AGGGTGGAAACCAGATGGCG	*crzA* with *AfupyrG* tail F
OHS360	CACCCATTGGTGTTGCGCTC	3′ flanking region of *crzA*
OHS361	GGGACGCGGTAGATTGTGCT	*crzA* nested 5′ NF
OHS362	TCGCACGCTTCCTTCAGGAG	*crzA* nested 5′ NR
OHS367	AATT **GCGGCCGC** GGGTTGGGATCCACCGCTTA	5′ *crzA* with pro and *Not*I
OHS368	AATT **GCGGCCGC** GGCATACCCCGAGTCCCCA	3′ *crzA* with *Not*I
OHS407	GATATGTCGCCACACTGGAC	*AflbrlA* F_RT
OHS408	CTGTATTCGCGGCTATTCGG	*AflbrlA* R_RT
OHS409	CTTCCGCACCTTAACAGCAG	*AflabaA* F_RT
OHS410	GTTTGCCGGAATTGCCAAAG	*AflabaA* R_RT
OHS411	CCGTCAGATATCCTGCCACA	*AflwetA* F_RT
OHS412	ATGGATATCGCGGGAGATGG	*AflwetA* R_RT
OHS405	TATGTCGGTGATGAGGCACA	*Aflactin* F_RT
OHS406	AACACGGAGCTCGTTGTAGA	*Aflactin* R_RT

^a^ Tail sequence is in italic. Restriction enzyme sites are in bold.

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
