# Peer review of "Function of crzA in Fungal Development and Aflatoxin Production in Aspergillus flavus"

_toxins, 2019, doi:10.3390/toxins11100567_

Round 1

Reviewer 1 Report

In this study the authors presented the characterization of the Aspergillus flavus CrzA gene encoding the main transcription factor of the calcineurin pathway. The authors generated CrzA deletion mutants of A. flavus and carried out a phenotypic analysis in comparison to the WT strain. The authors found reduced growth, reduced sclerotia production, reduced resistance to osmotic-induced stress compounds, and reduced production of aflatoxin.

A main question I have for the authors is whether they considered using a chemical inhibitor of calcineurin to see if they can recapitulate some of the phenotype of the CrzA deletion mutant. I looked on the web and it seems nobody did it, but I think it would be easy to treat A. flavus with tacrolimus or clyclosporin A and see whether the resistance to stresses or the mycotoxin production is similar to what is reported for the CrzA mutant. This would further prove the role of calcineurin in this biological processes. 

Beside this main point, I think this is a well written manuscript that is suited for publication in Toxins. I do think that there are few things that need attention before acceptance.

Probably figure 1 should contain an outgroup (for example a Penicillium species). Also I think that the character used to indicate the name species is a bit small. There are no indications on the tree generation in the material and method section.

Regarding the generation of mutants, I guess the strategy used is very common and there is not even need to show the mutant characterization by PCR?

Lines 103 – 111: this part is not clear and should be better written, and it should describe in better detail the related figure 2. For example there are no results describing figures 2D and 2E; it is not clear what is represented in the lower panels in figure 2A and B; in figure 2A (lower panels) and 2E is missing a scale bar.

Figure 3 is missing a statistical analysis. Still for RTqPCR, which one was the endogenous gene used for comparison? This has to be indicated in the text (it is missing also in M&M). In the M&M text related to the RTqPCR should be indicated how they excluded genomic contamination (no enzyme mix?) and how many biological replicated the authors performed.

Regarding the aflatoxin production, probably a HPLC analysis would have shown better whether crzA mutants produce aflatoxin. The TLC run did not seem very accurate so I am not sure how accurate is the quantification. In lines 163 there must be a typo, “whereas crzA strains DO NOT produce”

I think the discussion is a bit poor. From lines 197 to 206, and lines 207 to 217 seem more comments on negative results (shown in supplemental info) rather than a discussion. I suggest to move these parts in the result section, and write a better discussion on the interpretation of their results based on what is known in the field.

Reviewer 2 Report

The article is overall ok, but it does not really pop up for any new special information in the field. The authors simply repeated experiments that have been done in related species with the hope to find either similar or different results. No striking new information is provided.

Throughout the whole text they refer to mutant strains for the deletion mutant of crzA or the complemented strain. Is there a reason why multiple strains are involved? Generally the description consider one deletion and one complemented strain, regardless how many strains have been produced in the attempts.

Line 26: the "societal burden" is not exactly justified.

Line 35-41: there is no reason to describe those transcription factors in the introduction part. They are simply used as control for expression and can be briefly explained in the results part they are involved.

Line 50 and ahead: if it is possible to stay consistent with the gene denomination, crzA or crz1. Also when talking about other species, the use of two names generates more confusion here.

Many information provided about CrzA in other Aspergillus species are having the opposite effect than the one the author should aim for: the work presented here seem to lack completely in originality.

Line 85: which predicted CrzA?

Line 104-105: the name of the clones used in the lab are not relevant in the article; can be referred to simply as deletion mutant or complemented strain.

Line 120: "we asked" to who?

The aflatoxin paragraph is definitely too short, simply repeating experiment done in other species, it is not enough to justify its presence in the title and especially not the submission to a journal named Toxins.

Material and methods part is lacking details. For example, for the qRT-PCR, which methods have been used to calculate the significance? what about the statistic? The raw data have been submitted in accordance to the new regulation from 2015? What housekeeping gene have been used?

There is theb

Reviewer 3 Report

The authors have written a very interesting article about the function of CrzA, a transcription factor of the calcineurin pathway, in the filamentous fungus Aspergillus flavus. CrzA deletion mutants exhibited decreased colony growth, decreased number of conidia and decreased sclerocia production. Furthermore, mutant strains were more susceptible to osmotic pressure and cell wall stress than controls. Importantly aflatoxin production was significantly decreased in absence of crzA. I have some minor remarks listed below:

Introduction:

Line 24. Please change animal to animals.

Line 83. Please define AFLA_127920.

Results:

Figure 1. Please define NES and NLS in the figure legend

Figure 2. Line 107: Please define how the significance was calculated and indicate the significance in the figure (for example by “*”).

Please add incubation temperature.

Figure 4: Please indicate were the three stars “***” of line 143 belong in the figure and explain how the p value was calculated.

Figure 5: Please add incubation time and incubation temperature.

Line 163 Please add the word “not” after “delta crzA strains do”.

Figure 6: Please define the arrow in figure 6A.

Discussion:

Line 202: Please indicate that these results are not shown or show results in supplementary data.

Round 2

Reviewer 2 Report

The extensive modifications improved the quality of the paper.

I still believe in every publications more than a mutant and a complemented strains are produced and compared, still only one clone is presented in the paper and in the scientific community we simply refer to 1 single mutant strain, but this is a minor point.

As the authors acknowledge, the novelty of the paper is highly decreased by the fact that similar experiments have been performed in more than one Aspergillus species and their results simply confirm previous findings from other Aspergilli in A. flavus. Including further transcriptomic analyses and proteomic analyses, in order to provide a mechanism of action or at least some new information, would have been helpful to increase interest and value in this work. I understand the need of publishing results quickly, but still believe that for a good publication the work has to be complete and not partial. Maybe the authors can consider this point in the future.

Author Response

We really appreciate this valuable comments. Also we thank that you understand our status. Our next plan is that we will do RNA-seq and ChIP seq analyses to identify CrzA target genes. We will also conduct phospho-proteomic approch to verity CrzA is a bona fide calcineurin target in Aspergillus flavus. These results can make a complete story for the calcinsurin-CrzA cascade in A. flavus. 

Again, thank you for your meaningful advise.